# PlaceWaters: Real-time, explicit interface water sampling during Rosetta ligand docking

Shannon T. Smith[ID][1,2☯], Laura Shub[ID][3,4☯], Jens Meiler[ID][2,5,6]*

1 Chemical and Physical Biology Program, Vanderbilt University, Nashville, Tennessee, United States of America, 2 Center for Structural Biology, Vanderbilt University, Nashville, Tennessee, United States of America, 3 Biomedical Informatics Program, University of California San Francisco, San Francisco, California, United States of America, 4 Institute for Neurodegenerative Diseases, University of California San Francisco, San Francisco, California, United States of America, 5 Departments of Chemistry, Pharmacology, and Biomedical Informatics, Center for Structural Biology and Institute of Chemical Biology, Nashville, Tennessee, United States of America, 6 Institute for Drug Discovery, Leipzig University Medical School, SAC, Leipzig, Germany

☯ These authors contributed equally to this work.
* jens@meilerlab.org

**Data Availability Statement:** All relevant data are within the paper and its Supporting information files.

**Funding:** STS received funding for this work through the National Cancer Institute of the

## Abstract

Water molecules at the protein-small molecule interface often form hydrogen bonds with both the small molecule ligand and the protein, affecting the structural integrity and energetics of a binding event. The inclusion of these 'bridging waters' has been shown to improve the accuracy of predicted docked structures; however, due to increased computational costs, this step is typically omitted in ligand docking simulations. In this study, we introduce a resource-efficient, Rosetta-based protocol named "PlaceWaters" to predict the location of explicit interface bridging waters during a ligand docking simulation. In contrast to other explicit water methods, this protocol is independent of knowledge of number and location of crystallographic waters in homologous structures. We test this method on a diverse protein-small molecule benchmark set in comparison to other Rosetta-based protocols. Our results suggest that this coarse-grained, structure-based approach quickly and accurately predicts the location of bridging waters, improving our ability to computationally screen drug candidates.

## Introduction

### Computational ligand docking in drug discovery

Structure-based computer-aided drug discovery (SB-CADD) is a field at the intersection of computer science and structural biology that uses computational tools to identify small molecule (ligand) binders from the three-dimensional structure of a protein [1]. Computational protein-ligand docking attempts to predict the structure of a small molecule ligand in complex with its associated protein. Examples of docking programs include RosettaLigand [2–5], Glide [6], and AutoDock [7]. This protocol will focus on RosettaLigand, a ligand docking protocol within Rosetta that takes advantage of the Rosetta protein modeling infrastructure and scoring

National Institutes of Health (F31 CA243353). The funders had no role in study design, data collection and analysis, decision to publish, or preparation of the manuscript. STS also received funding through the PhRMA Foundation's Pre-Doctoral Fellowship in Informatics (phrmafoundation.org). LS did this work as an intern in the Rosetta REU Program, which is funded by the NSF Div Of Biological Infrastructure (Award Number: 1659649).

**Competing interests:** The authors have declared that no competing interests exist.

function to virtually dock small molecule ligands with full backbone, side chain, and ligand flexibility.

## Protein-ligand interface water molecules play a crucial role in binding

Up to two-thirds of protein-ligand interfaces contain one or more water molecules within the binding site, many of which form hydrogen bond networks with both the ligand and the protein [8]. These "bridging waters" play a key structural role in the binding of a ligand within an active site, and in many cases dictate the geometry and energetics of a binding event. One well-studied example is HIV-protease, where key water molecules in the interface stabilize the binding of class HIV-1 protease inhibitors [9, 10]. Despite this, water is typically ignored during protein-ligand docking studies due to the increased computational complexity introduced by both predicting and modeling these waters.

## Ligand docking with explicit waters improves accuracy

Previous studies in other common docking software show that including explicit waters in the immediate vicinity of both the protein and the ligand improves overall docking accuracy [6, 7, 11, 12]. Past efforts to model the effects of water in RosettaLigand leverage the ability of Rosetta to simultaneously dock multiple small molecules such as waters and cofactors, either keeping waters stationary relative to the protein ("protein-centric") or translating waters along with the ligand ("ligand-centric") [13]. These approaches have limited applicability, since they rely on the location and number of experimentally-determined water locations *a priori*. Recent parallel efforts within Rosetta use both implicit and explicit water representation methods to improve sampling and scoring terms for coordinated water molecules at protein-protein and protein-small molecule interfaces [14–16]. The method described herein complements these approaches within Rosetta, in particular allowing for rapid placement of water molecules for later refinement.

## Prediction of water molecules in protein-small molecule interfaces continues to be a largely unanswered question in the field

There have been many approaches to this problem using both explicit and implicit water molecule representation, including (but not limited to) one or a combination of molecule mechanics force fields [15–19], knowledge-based terms [18, 20, 21], homology structures [22], and manual curation. To the authors knowledge, the 2015 CAPRI assessment [23] is the most recent blind prediction specifically looking at water molecules in a single protein-protein interface example, and the results highlight the difficulty of this problem. Briefly, teams were first asked to predict a protein-protein interaction from a highly conserved system then asked to predict the water network at this interface. Most teams successfully predicted the protein-protein interaction; however, water predictions were largely unsuccessful with only 6% of models successfully recaptured >50% of water-mediated contacts. This type of assessment has not been performed since, nor specifically for protein-small molecule interactions.

Here we describe the "PlaceWaters" protocol, a new algorithm to predict the locations of bridging waters during a docking simulation. Notably, and in contrast to previously described methods, PlaceWaters is independent of prior knowledge of the location of waters in experimental structures or the number of waters to place within a binding site. We also present the results on a diverse benchmark of protein-ligand associations. Our results demonstrate that our structure-based water placement algorithm can quickly and correctly predict the locations of bridging waters to the same success as previously described methods. Though the results are

promising, we also offer areas for potential improvement that build on our method described here.

## Materials and methods

### The PlaceWaters mover algorithm

First, a grid is superimposed onto the ligand, where size of the grid is double the distance between the center of the ligand to the most distal heavy atom. We iterate through all grid voxels and record unoccupied coordinates. We further prune this set of "candidate coordinates" to remove all coordinates below a certain cutoff to existing atoms, leaving only coordinates that are a sufficient distance from both the ligand and the protein. The remaining coordinates are clustered into "voids" based on a user-defined distance and any clusters that are within 2Å of each other are merged. The final set of candidate coordinates is the centroid of each resulting cluster. The final step in the algorithm involves checking each of the remaining candidate coordinates for hydrogen bond donors and acceptors in a shell around them. If a hydrogen donor or acceptor atom is present on both the ligand and the protein within the desired cutoff values, the coordinate is accepted, and a water oxygen is placed at this coordinate. These steps are represented visually in Fig 1.

Parameter optimization to set default cutoff distances for merging, clustering, and accepting possible hydrogen bonds was performed using Bayesian optimization in the 'hyperopt' software [24]These values can be easily modified to the user's preference in the XML script

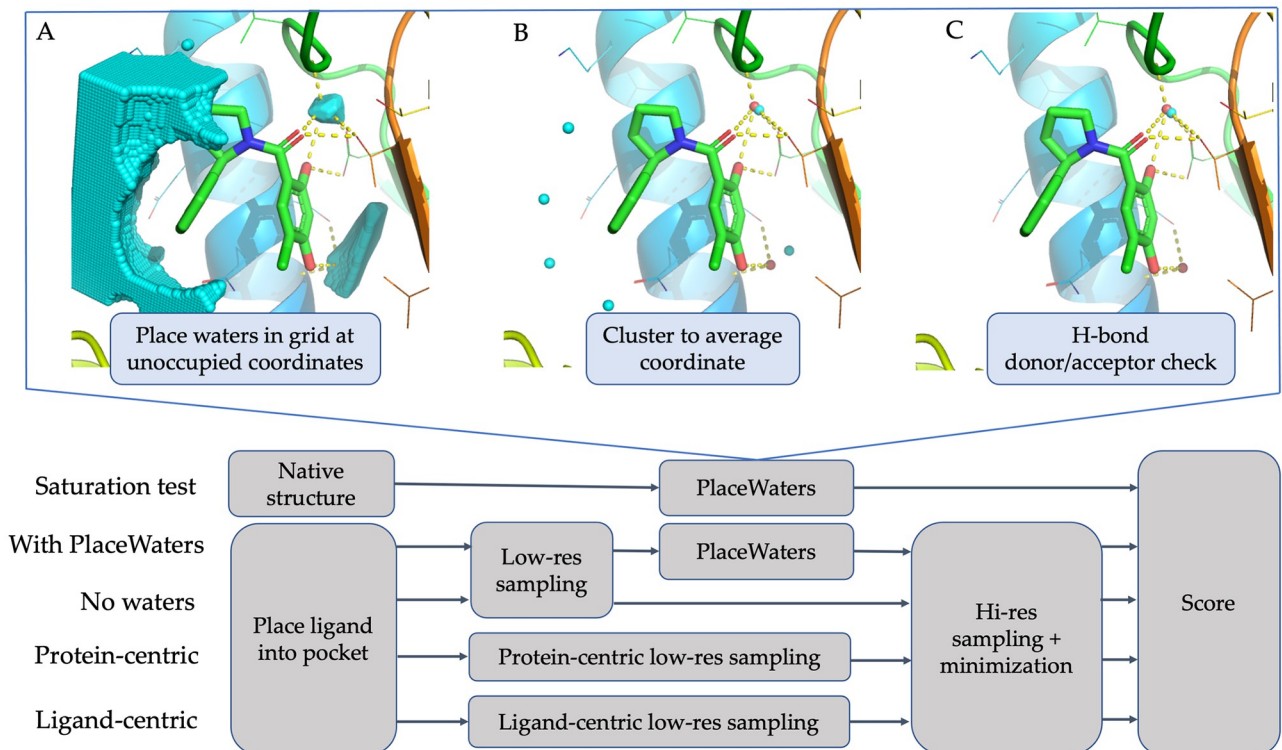

**Fig 1. Details on PlaceWaters algorithm and comparison to all tested protocols.** Top: PlaceWaters steps A) water grid at given density around ligand in unoccupied coordinates; B) cluster waters to average coordinate; C) final water placement after hydrogen bonding donor or acceptor check. Co-crystallized waters are shown in red spheres, predicted waters depicted in cyan spheres (PDB: 3K97). Bottom: Workflows for each protocol starting with the "saturation test" along the top row followed by the different docking tests "With PlaceWaters", "No waters", "Protein-centric", and "Ligand-centric".

(see S1 File). We use the RMSD distance between oxygen molecules in equivalent native and added waters as the objective function during optimization. The input files used were the native poses of the training associations, stripped of all water molecules. The resulting optimal values of each parameter were used to perform the benchmark test.

## Saturation test

The "saturation test" looks to determine the ability of the PlaceWaters algorithm to put waters correctly into the native protein-ligand interface. This takes the native protein-ligand interface and places the waters accordingly without any structural perturbation of the protein or the ligand. We break down the algorithm performance to determine the loss of correct water placement at each step. In each case, we calculate the distance between the oxygen atoms in each bridging water and the nearest placed water molecule. A success case is classified as a water is placed within 1.4Å of a bridging water. Separately, we look at the number of waters added compared to the known native waters for over- and under-estimation of water addition.

## Docking tests

Inclusion of water placement is meant to be used to improve ligand docking predictions, therefore we ran the benchmark set against other Rosetta-based protocols for comparison. The relationship between these protocols is displayed in Fig 2. In each docking test (RosettaLigand with PlaceWaters, RosettaLigand without PlaceWaters, and RosettaLigand with protein- and ligand- centric water placement), we used the same metrics as in the saturation test to determine successful water placement. Additionally, we wanted to know if the addition of explicit water molecules helped with ligand binding predictions during docking. To determine if

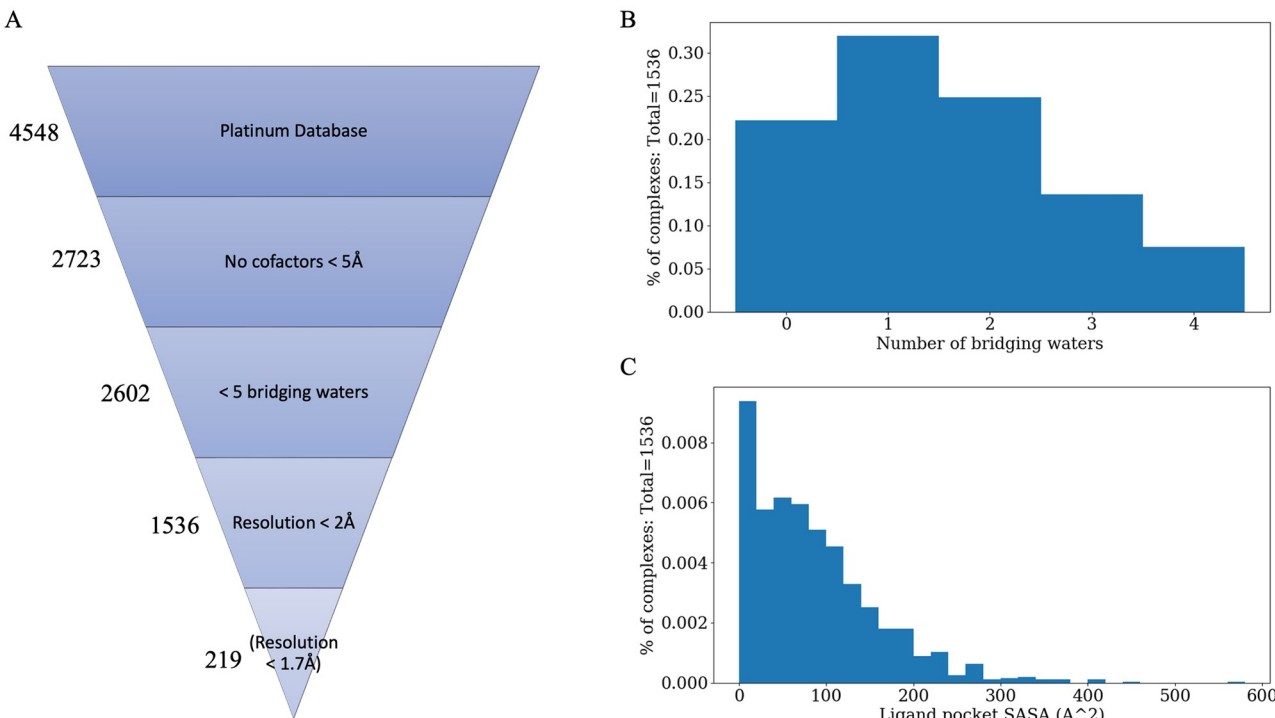

**Fig 2. Dataset information.** A) Selection criteria with respective number of complexes that pass; Distribution of B) number of bridging waters and C) ligand pocket solvent-accessible surface area across dataset.

PlaceWaters aided in sampling the native ligand binding pose, we simply compare the percentage of output models <2Å RMSD of the native binding mode between RosettaLigand with and without PlaceWaters. We also wanted to know if the inclusion of explicit waters scores native-like ligand poses more favorably. To determine this, we compare the number of <2Å models in the top 10% by score between RosettaLigand with and without PlaceWaters.

## RosettaLigand with and without PlaceWaters

The standard ligand docking protocol in Rosetta briefly consists of the following steps: initial placement of the ligand into the binding pocket, low- then high-resolution sampling of the ligand, and a final energy minimization. Using the average coordinate of the co-crystallized ligand as a starting point, the low-resolution step undergoes random transformations, with an initial perturbation of 3.0 Å followed by random perturbations within 5 Å and 5˚, or a switch to a different ligand conformer. This occurs 500 times through a Monte Carlo sampling process, wherein a favorable change is always accepted, and an unfavorable change is accepted based on the Metropolis criteria. The resulting lowest-energy pose is the final output. In high-resolution sampling, the ligand, any present waters, and side chains within 6.0 Å of the ligand or within 2.0 Å of any waters, are optimized by small translations up to 0.1 Å and rotations of up to 5˚. This is followed by a gradient descent algorithm that further adjusts the ligand and side chain positions. This process is repeated six times, again utilizing a Monte Carlo approach as described above. The high-resolution step is followed by the final minimization step in which all waters, ligand torsion angles, as well as backbone φ and ψ angles and side chain χ angles within 7.0 Å of the ligand or 2.5 Å of any water molecules are perturbed to find the local energetic minimum. The final output pose is scored and evaluated.

Importantly, we excluded potential differences derived from the low-resolution sampling step by using the same output from the low-resolution steps in both RosettaLigand protocols with and without the PlaceWaters mover. In each protein-ligand case, we produced 1000 outputs from low-resolution sampling to then be used as input for the PlaceWaters step (RosettaLigand with PlaceWaters) or straight into high-resolution docking steps (RosettaLigand without PlaceWaters).

## RosettaLigand with protein- and ligand-centric waters

We also compared performance against the existing protein- and ligand-centric docking protocols [11], using the protein with the interface bridging waters already present as the starting protein pose for docking. The protein-centric docking method places water molecules in their known crystallographic coordinates and subsequently performs low-resolution ligand docking while translating these waters simultaneously with respect to protein structure perturbations. The ligand-centric docking method also uses known crystallographic water coordinates, but the waters are translated along with the ligand coordinates during the low-resolution sampling step.

## Benchmark dataset preparation

Selection and curation of a diverse small molecule dataset is visually represented in Fig 3. The benchmark proteins are taken from the set of associations known as the "Platinum set" [25] and the CSAR 2014 benchmark [26]. The original dataset contained 4548 protein-ligand interactions; however, we removed systems containing DNA/RNA, non-water cofactors within 6Å of the ligand, more than 6 bridging waters, and greater than 1.7Å structure resolution. The final dataset contained 219 protein-ligand interactions. "Bridging waters" were defined as those within 3Å of both the ligand and the protein in the native structure while "local waters"

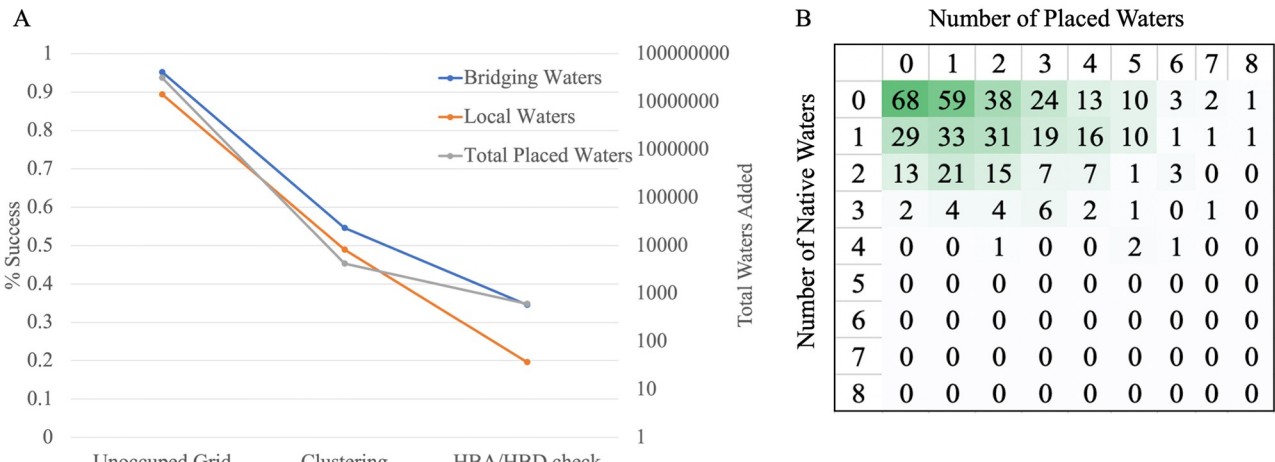

**Fig 3. Saturation test results.** A) Blue: percentage of crystallographic waters that contain a placed water within 1.4Å at each filtering step; orange: Total number of placed waters in dataset after each filtering step. B) Comparison of number of crystallographic waters versus number placed in dataset.

were defined as those within 5Å of any ligand atom (see get_native_waters.py script in S1 File). The total dataset of 219 protein-ligand complexes contains a combined 480 and 2268 bridging waters and local waters, respectively. The PlaceWaters protocol is designed to recapture the bridging waters between a protein and ligand, but we also include local waters for evaluation.

# Results

## The saturation test demonstrated that PlaceWaters successfully predicts explicit waters at the binding interface

We initially tested performance on the native crystallographic ligand poses of the benchmark set to determine how well the PlaceWaters mover determines water locations without protein and ligand docking. Out of 480 bridging water molecules across 219 complexes, we successfully recapture 166 bridging waters (35%). While this percentage appears low, it is on par or better when compared to competing algorithms with the same task while other algorithms have not been benchmarked on this blind prediction. For comparison, the overall results of the CAPRI assessment demonstrated that out of 176 high or medium quality docking models, 44% successfully recaptured 30% of water molecules at the protein-protein interface, and 6% captured greater than 50%. Another key finding in the CAPRI assessment demonstrated that even the successful docking models overestimate the number of waters resulting in roughly half of water contacts as false positives. Similarly, the PlaceWaters mover overestimates the number of waters in its predictions (Fig 4). Over-addition could be corrected by adjusting several parameters: decreasing the allowed cutoffs for acceptable hydrogen bond donors and acceptors, leading to fewer candidate meeting these criteria or increasing the clustering distance when selecting candidate coordinates, leading to fewer, larger clusters. Since we want to predict putative water locations for further refinement, we aired on the side of over-predictions. Increasing the allowed hydrogen bond donor/acceptor distance or lowering the clustering distance cutoffs too much, however, tends to over-predict waters in the pocket. In a few cases, even using the optimized cutoffs, PlaceWaters adds up to eight water molecules in a pocket without any bridging waters. Upon inspection of these individual cases, these ligand pockets are entirely solvent-exposed and likely contain bulk water in these regions.

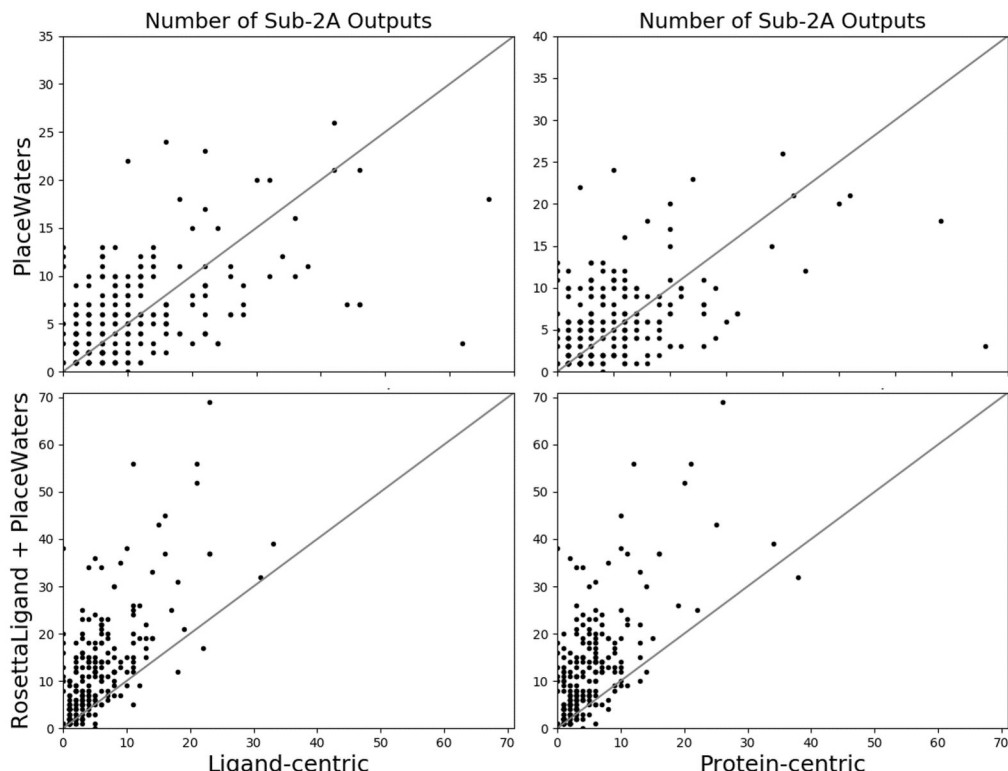

**Fig 4. Number of sub-2Å ligand RMSD output models in comparison to ligand-centric (left) and protein-centric (right) methods.** Within the top 5% by interface score.

### The full RosettaLigand protocol with the PlaceWaters mover does not improve docking results in comparison to omitting the PlaceWaters step

We first determined if the addition of PlaceWaters increases sampling the native ligand binding pose by comparing the number of sub-2Å outputs when using RosettaLigand with and without the PlaceWaters step. Based on this test, there fails to be an improvement in sampling low-RMSD binding poses when using PlaceWaters (Fig 4). This result is expected based on the order of steps in the pipeline and PlaceWaters being implemented between the low- and high-resolution sampling steps. Typically, the ability to sample near-native ligand conformations is based on the low-resolution step where there are larger ligand perturbations throughout the pocket.

We then tested if adding the PlaceWaters step enriched near-native ligand poses by comparison the number of near native ligand binding poses within the top 5% of output interface scores. Based on our benchmark set, there is also little improvement in how scoring with and without water placement (Fig 5). Since we have not refactored any of the scoring methods to include water-based interactions, this is expected.

### PlaceWaters algorithm improves docking results in comparison to protein- and ligand-centric water placement algorithms

The addition of PlaceWaters into the RosettaLigand protocol, however, did produce more sub-2Å ligand RMSD outputs in comparison to the other explicit water protocols in almost all cases. By breaking these results down by steps in the protocol, this improvement seems to

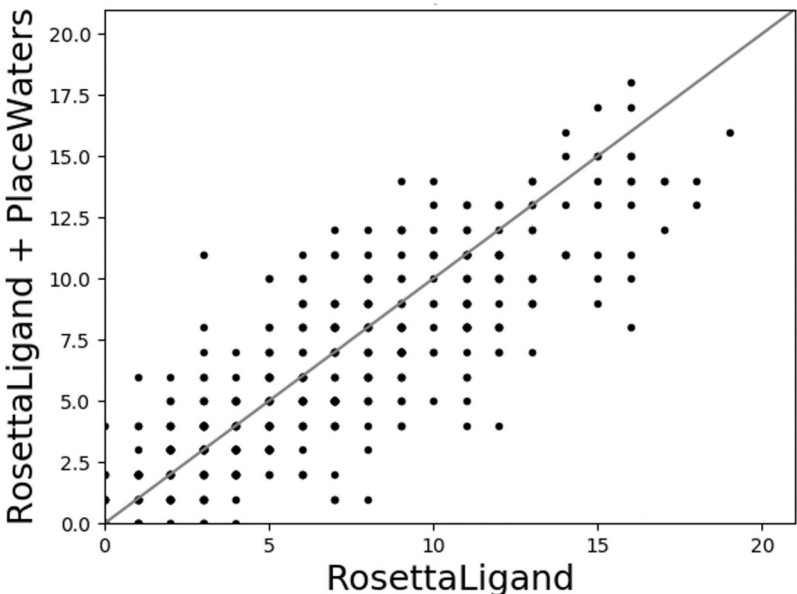

**Fig 5. Number of sub-2Å ligand RMSD output models within the top 5% by interface score.**

come from the post-PlaceWaters steps, namely the high-resolution sampling and minimization steps (Fig 5). Although this isn't the PlaceWaters mover enabling this improvement in itself, these findings suggest that the inclusion of explicit waters in the refinement step improves sampling accuracy.

## Runtime of PlaceWaters mover

In adding to the full RosettaLigand docking protocol, it is important that any additional steps are not prohibitively time consuming. In comparison to other resource-intensive explicit water representation algorithms, the PlaceWaters mover took an average of 6 seconds to run across our dataset. This is faster than the high-resolution dock which takes an average of 9 seconds, but slower than the low resolution, side chain repacking, and final minimization, which each had runtimes less than 2 seconds. This runtime indicates that addition of the PlaceWaters mover to the overall docking protocol keeps the overall runtime of each trial short, making this method possible to include in more extensive docking simulations including virtual screening.

## Case-study: Heat shock protein 90 in complex with known inhibitor

For a detailed walk-through of the algorithm, we present a case-study outlining the results from the co-crystal structure of heat shock protein 90 (Hsp90) in complex with a known inhibitor, 4-chloro-6-{[(2R)-2-(2-methylphenyl)pyrrolidin-1-yl]carbonyl}benzene-1,3-diol (PDB ID: 3K97, ligand ID: 4CD) [27]. As can be seen in Fig 1, there are 2 co-crystallized bridging waters: HOH 237, which forms hydrogen bonds with the ligand para-hydroxyl, the L63 backbone carbonyl oxygen and the S67 sidechain hydroxyl group; and HOH 238, which forms hydrogen bonds with the ligand ortho-hydroxyl and ligand amide oxygen, D108 sidechain carboxyl group and the G112 backbone amide nitrogen. All numbering is based on the deposited PDB 3K97.

As can be seen in Fig 1A, the initial grid successfully identifies the regions where HOH 237 and 238 were co-crystallized, as well as a large region on the solvent-facing side of the ligand. Fig 1B shows the results from the clustering and merging step resulting in 6 waters being placed around the ligand, including waters placed at 1.2Å and 0.3Å of HOH 237 and 238, respectively. As shown in Fig 1C, the PlaceWaters mover kept the water molecule near the HOH 238, and successfully recaptured this hydrogen bond network. The water placed close to HOH 237; however, was removed during the final selection step due to being too far from either a ligand or protein hydrogen bond donor or acceptor atom.

## Discussion

### The PlaceWaters mover successfully predicts explicit water molecule placement at native protein-ligand interfaces

The saturation test demonstrates the ability of PlaceWaters to successfully place water molecules within the protein-ligand pocket. This method is strictly statistics-based and is independent of time-consuming calculations to optimize water placement. This algorithm is also independent of *a priori* knowledge on the number of waters to add in the pocket, and cutoff values can be tuned to add more or fewer waters depending on different factors such as the ligand, pocket accessibility, etc. This is optimistic for future work into using explicit water representation within the Rosetta framework, and could be used in tandem with recent work done elsewhere in the community [14].

### Over-prediction of water molecules in interfaces

The function used to calculate water RMSD did not penalize false positives as it only considered waters that it was able to pair with the closest native crystallographic equivalent. As this same function was used as the objective function in parameter optimization, this resulted in the addition of a greater number of waters than were present in the native structure according to the interface water criteria. This is especially true for associations with larger binding pockets that result in a large number of candidate clusters and ligands with many hydrogen donors and acceptors. This setup was used intentionally in order to explore all possible locations for water molecules. However, this bias could be corrected by limiting the hydrogen contact cutoff values to accept a narrower set of candidates. The PlaceWaters mover allows for user-modified parameters as desired. We also felt that the over-prediction of water is better than under-prediction in that outputs can be manually, albeit tediously, curated post-simulation to filter out unlikely or undesired water placements based on specific residues or substituent groups of interest.

### Further parameter optimizations

While the PlaceWaters algorithm correctly predicts the location of bridging waters in certain trials, the average number of waters added for each trial is consistently around three, regardless of the number of native crystallographic bridging waters, and can most likely be attributed to the cutoff values. As seen in the 3K97 case-study, the final selection step removed a bridging water molecule near HOH 237 due to being slightly too far from a hydrogen bond donor or acceptor atom. In this case, it would be better to tune these distance settings. Another potential way to improve this is to do a more fine-grained sampling of the water within a defined sphere to check for nearby hydrogen bond acceptor and donor atoms. This could avoid such sharp cutoffs and be less reliant on the coarse-grained merged coordinate represented. These values were optimized on the training set using the hyperopt hyperparameter software with an

objective function that did not penalize false positives. It is likely that optimizing these cutoff values using an alternative objective function would result in different cutoff values. Exploring this in the future would result in both more accurate water prediction as well as more trials with the correct number of waters added.

### Interface water networks and protein-protein interactions

In addition to directly bridging waters that form hydrogen bonds with the ligand and the protein, many protein-ligand interfaces contain a network of water-water hydrogens bonds that also contribute to the native docked structure of the ligand. The PlaceWaters algorithm could potentially be updated to predict these networks by expanding the criteria for placing a water at specific coordinates to include cases where other water candidate coordinates are within this water-water hydrogen bonding distance. Similarly, bridging waters contribute to protein-protein docking, and an algorithm to predict the location of bridging waters in these docks could lead to improved structure prediction in these cases, as well. Future modifications to the PlaceWaters mover could further generalize the algorithm to apply to these docks by modifying the initial grid placement to encompass the protein-protein interface rather than the ligand binding pocket.

## Conclusion

PlaceWaters achieves successful water placement (average water RMSD below 1.4 Å) in a large number of protein-ligand associations tested. The addition of PlaceWaters improved ligand docking sampling over other Rosetta-based methods that use explicit water representation. Finally, incorporation of the PlaceWaters mover to the full docking protocol extended the duration of each trial by an average of approximately 6 seconds, allowing it to be incorporated into the docking protocol without extending the total time of each trial to unviable lengths. The results of this benchmark serve as evidence that structure-based water prediction algorithms are able to predict the locations of bridging waters in ligand docking with relatively low runtimes and provides a basis for further exploration into incorporating bridging waters in future computational ligand docking efforts, particularly in cases with buried, medium-sized binding pockets.

## Supporting information

**S1 File. Protocol capture.** All scripts for dataset curation, running tests and analysis provided here.
(GZ)

## Acknowledgments

The authors would like to thank members of the RosettaCommons community for feedback on method and code development, as well as the Rosetta REU program through Johns Hopkins University and Dr. Jeffrey Gray.

## Author Contributions

**Conceptualization:** Shannon T. Smith, Laura Shub, Jens Meiler.

**Data curation:** Shannon T. Smith.

**Formal analysis:** Shannon T. Smith.

**Investigation:** Shannon T. Smith, Laura Shub.

**Methodology:** Laura Shub.

**Software:** Laura Shub.

**Supervision:** Jens Meiler.

**Writing – original draft:** Shannon T. Smith.

**Writing – review & editing:** Laura Shub, Jens Meiler.

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
