## [Decision Letter · Decision Letter 0]

30 Mar 2022

PONE-D-22-03327PlaceWaters: real-time, explicit interface water sampling during Rosetta ligand dockingPLOS ONE

Dear Dr. Smith,

Thank you for submitting your manuscript to PLOS ONE. After careful consideration, we feel that it has merit but does not fully meet PLOS ONE’s publication criteria as it currently stands. Therefore, we invite you to submit a revised version of the manuscript that addresses the points raised during the review process.

We look forward to receiving your revised manuscript.

Kind regards,

L. Michel Espinoza-Fonseca

Academic Editor

PLOS ONE

Journal Requirements:

2. We note that you have referenced (Carlson H, Smith R, Damm-Ganamet† K, Stuckey J, Ahmed A, Convery M. CSAR 2014: A Benchmark Exercise Using Unpublished Data from Pharma. Journal of Chemical Information and Modeling. 2016:56(6):1063–7) which has currently not yet been accepted for publication. Please remove this from your References and amend this to state in the body of your manuscript: (Carlson H, Smith R, Damm-Ganamet† K, Stuckey J, Ahmed A, Convery M. CSAR 2014: A Benchmark Exercise Using Unpublished Data from Pharma. Journal of Chemical Information and Modeling. 2016:56(6):1063–7 [Unpublished]”) as detailed online in our guide for authors

Reviewers' comments:

Reviewer's Responses to Questions

**Comments to the Author**

1. Is the manuscript technically sound, and do the data support the conclusions?

Reviewer #1: Yes

2. Has the statistical analysis been performed appropriately and rigorously? 

Reviewer #1: Yes

3. Have the authors made all data underlying the findings in their manuscript fully available?

Reviewer #1: Yes

4. Is the manuscript presented in an intelligible fashion and written in standard English?

Reviewer #1: Yes

5. Review Comments to the Author

Reviewer #1: In this work, the authors introduced a new algorithm to predict the location of bridging waters during molecular docking. This approach is quite exciting and provides relevant information to this highly complex topic. I have just a few comments and suggestions that I would like the authors to address.

- The images in Figure 1 do not provide enough information on their own about the methodology. In addition, the predicted bridging waters (small gray asterisks) are not visible, making it difficult to compare them with the structural water molecules (red spheres). From my point of view, the authors should merge Figure 1 with Figure 2, explaining in more detail the steps of adding bridging waters so that the reader can have a better appreciation of the algorithm.

- I think it is essential to include a case study. It could be the one used in the "provided_protocol" directory.

- Page 17, Line 226. Reference error.

- When I tried to use the provided protocol scripts, I ran into some issues:

(1) Authors should specify the packages from the beginning that should be installed, i.e., Corina, OpenBabel, BCL (Biology and Chemistry Library Project), and Rosetta.

(2) Is there any additional option to prepare the 3D structures of the ligands instead of Corina? In my case I used OpenBabel (babel -isdf ${prefix}.native.sdf -osdf ${prefix}.corina.sdf --gen3D).

(3) There seems to be an error in the PARAMS file. The 3-letter code should be 4CD instead of 3K9; otherwise, the program throws an error.

(4) Will a particular version or plug-in of Rosetta should be installed/included to run this module? I tried to run it with my actual version and got the following error: "Element 'PlaceWaters': This element is not expected". I think it's important to let the reader know where they can find this information (e.g., github).

6. PLOS authors have the option to publish the peer review history of their article (what does this mean?). If published, this will include your full peer review and any attached files.

Reviewer #1: No

---

## [Author Response · Author response to Decision Letter 0]

19 Apr 2022

We want to thank the Reviewers for their input to improve this paper. Please see attached Response To Reviewers file for point-by-point changes to the manuscript.

---

## [Editor Report · Decision Letter 1]

16 May 2022

PlaceWaters: real-time, explicit interface water sampling during Rosetta ligand docking

PONE-D-22-03327R1

Dear Dr. Smith,

We’re pleased to inform you that your manuscript has been judged scientifically suitable for publication and will be formally accepted for publication once it meets all outstanding technical requirements.

Kind regards,

L. Michel Espinoza-Fonseca

Academic Editor

PLOS ONE

---

## [Editor Report · Acceptance letter]

20 May 2022

PONE-D-22-03327R1 

PlaceWaters: real-time, explicit interface water sampling during Rosetta ligand docking 

Dear Dr. Smith:

I'm pleased to inform you that your manuscript has been deemed suitable for publication in PLOS ONE. Congratulations! Your manuscript is now with our production department. 

Kind regards, 

on behalf of

Dr. L. Michel Espinoza-Fonseca 

Academic Editor

PLOS ONE